# Prescription Trends of Biologic DMARDs in Treating Rheumatologic Diseases: Changes of Medication Availability in COVID-19

**DOI:** 10.3390/jpm13081199

**Published:** 2023-07-28

**Authors:** Mislav Radić, Hana Đogaš, Karla Vrkić, Andrea Gelemanović, Ivanka Marinović, Dijana Perković, Jurica Nazlić, Josipa Radić, Daniela Marasović Krstulović, Julije Meštrović

**Affiliations:** 1Internal Medicine Department, Rheumatology, Allergology, and Clinical Immunology Division, University Hospital of Split, 21000 Split, Croatia; 2Department of Internal Medicine, University of Split School of Medicine, 21000 Split, Croatia; 3Internal Medicine Department, Nephrology and Haemodialysis Division, University Hospital of Split, 21000 Split, Croatia; 4Mediterranean Institute for Life Sciences (MedILS), 21000 Split, Croatia; 5Physical Medicine and Rehabilitation with Rheumatology Division, University Hospital of Split, 21000 Split, Croatia; 6University Department of Health Studies, University of Split, 21000 Split, Croatia; 7Internal Medicine Department, Emergency Medicine, Intensive Care, and Clinical Pharmacology with Toxicology Division, University Hospital of Split, 21000 Split, Croatia; 8Paediatric Diseases Department, University Hospital of Split, 21000 Split, Croatia; 9Department of Paediatrics, University of Split School of Medicine, 21000 Split, Croatia

**Keywords:** rheumatologic diseases, biologic DMARDs, COVID-19

## Abstract

Biologic disease-modifying antirheumatic drugs (DMARDs) are very effective in treating rheumatic diseases with a good patient tolerance. However, high cost and individualistic approach requires dedication of the physician. Therefore, the aim of this study was to determine how the COVID-19 pandemic has affected the prescription of biologic DMARDs in rheumatology at the University Hospital of Split. The data collection was conducted through an archive search in the Outpatient Clinic for Rheumatology in the University Hospital of Split, Split, Croatia. The search included the period before and after the start of the COVID-19 pandemic in Croatia (31 March 2020). Collected data included age, sex, ICD-10 code of diagnosis, generic and brand name of the prescribed drug, date of therapy initiation, and medication administration route. In the pre-COVID-19 period, 209 patients were processed, while in the COVID-19 period, 185 patients were processed (11.5% fewer). During pre-COVID-19, 231 biologic medications were prescribed, while during COVID-19, 204. During COVID-19, IL-6 inhibitors were less prescribed (48 (21%) vs. 21 (10%) prescriptions, *p* = 0.003), while IL-17A inhibitors were more prescribed (39 (17%) vs. 61 (30%) prescriptions, *p* = 0.001). In ankylosing spondylitis (AS), adalimumab was prescribed more during pre-COVID-19 (25 vs. 15 patients, *p* = 0.010), while ixekizumab was prescribed less (1 vs. 10 patients, *p* = 0.009). In rheumatoid arthritis, tocilizumab was prescribed more in the pre-COVID-19 period (34 vs. 10 patients, *p* = 0.012). Overall, the prescription trends of biologic DMARDs for rheumatologic diseases did not vary significantly in the University Hospital of Split, Croatia. Tocilizumab was prescribed less during COVID-19 due to shortages, while ixekizumab was more prescribed during COVID-19 due to an increase in psoriatic arthritis patients processed and due to being approved for treating AS.

## 1. Introduction

The novel coronavirus called Severe Acute Respiratory Syndrome Coronavirus 2 (SARS-CoV-2) is a highly infectious coronavirus that emerged in late 2019 and caused a pandemic of acute respiratory illness: Coronavirus Disease 2019 (COVID-19) [1]. The outbreak was a global event that prompted the World Health Organization to declare a pandemic on 11 March 2020 [2]. The spread of the virus led to an overload of the public health system and forced the application of measures to prevent and contain the spread of the virus, including quarantine of infected individuals, reduction of social contact, self-isolation, avoidance of public gatherings, frequent disinfection, and wearing of a mask [3]. The organization of the health system had to adapt to the needs of treating patients with COVID-19 by establishing isolation units within hospitals and reducing regular activities to protect patients and prevent spread [4]. Therefore, staffing schedules were reorganized and divided into two groups, each working 2-week shifts to ensure that backup staff were available for potentially infected staff. The introduction of shift work resulted in a reduction in hospital outpatient consultations, potentially affecting the level and quality of care for patients with chronic diseases and the availability of physicians [5].

Biologic disease-modifying antirheumatic drugs (DMARDs) are an important and innovative therapeutic option that specifically target molecules involved in the development of autoimmune diseases and provide an alternative to existing treatments: antirheumatic drugs and other immunosuppressive drugs normally used to treat autoimmune diseases [6]. Cytokines (tumor necrosis factor α (TNF-α), various interleukins), activated T cells, and B cells are associated with the development of rheumatic diseases and provide a target for biologic therapy [7]. Biologics have a good overall efficacy and safety profile, and these targeted therapies are often well tolerated by patients [6]. Some of the drugs approved for rheumatoid arthritis, psoriatic arthritis, and ankylosing spondylitis are TNF-α inhibitors (adalimumab, certolizumab, golimumab, infliximab, and etanercept) and have been shown to be good therapeutic choices [8]. However, disadvantages of biologics include inefficacy of intravenous use, high cost, and the fact that some patients exhibit primary nonresponse or secondary loss of response to biologics [6,9]. Therefore, the decision of initiation of biologic DMARDs requires approval and a consensus of the board.

It is important to note that the University Hospital of Split is the only tertiary center in Split-Dalmatia County and in a wider southern region of Croatia with a population of approximately one million people. In addition, the primary site for rheumatology care was repurposed multiple times during the COVID-19 period to accommodate infectious patients and critical care. In light of these challenges, a “virtual outpatient clinic” was established at the University Hospital of Split to maintain the level and standard of care for these patients and to facilitate physician consultation. The dynamics of treatment with biologic DMARDs according to clinical response was performed according to The European Alliance of Associations for Rheumatology (EULAR) recommendations [10].

One of the main problems reported by patients during similar virtual visits organized during the COVID-19 pandemic was the lack of face-to-face communication and clinical examination, with most preferring a face-to-face visit or at least video communication [10]. In another study, patients indicated that interactions during a face-to-face visits were of high personal importance [11]. Another study that examined patient satisfaction and experiences with virtual visits found that technical difficulties were associated with lower overall satisfaction [12]. Patients’ willingness to use technology and comfort with the technology played an important role in their experience and in the overall treatment experience, which influenced patient–physician interaction [12]. Therefore, the quality of patient–physician interaction can potentially influence decision making.

In addition, the COVID-19 pandemic challenged the decision-making process and treatment with biologic DMARDs because the infection could potentially influence the primary autoimmune disease [13]. In France, a sharp increase in new prescriptions of hydroxychloroquine and tocilizumab was noted, while other biologic agents, methotrexate and apremilast, were prescribed less frequently [14]. A study conducted in the United States by Mehta et al. also found that the number of patients referring to their rheumatologist varied, and that rheumatologists in areas with high COVID-19 caseloads, compared with areas with low caseloads, had different medication management [15]. Overall, by studying all of these changes, we can analyze how potential medication shortages, lack of face-to-face visits, physician availability, and virtual consultations have affected the care and treatment of these patients. Therefore, the aim of this study was to determine how the COVID-19 pandemic has affected the prescription of biologic DMARDs for rheumatologic diseases at the University Hospital of Split, to identify the market trends of specific drugs, and to discuss how this might affect the quality of care.

## 2. Materials and Methods

### 2.1. Study Population and Design

This cross-sectional study, aimed at investigating the prescribing patterns of biological DMARDs before and after the official onset of the COVID-19 pandemic, was conducted at the Outpatient Clinic for Rheumatology, University Hospital of Split, Croatia. Patients treated in the clinic for rheumatologic diseases under different International Classification of Diseases, Tenth Revision (ICD-10) codes were included in the study [16]. Data collection was performed through an archival search between January 2022 and March 2022. The data search covered the period of 21 months before and after the declaration of the official start of the pandemic COVID-19 in Croatia (31 March 2020), i.e., the collected data included records from 1 July 2018 to 21 December 2021. The collected data included age, sex, ICD-10 code of diagnosis, generic and brand name of the prescribed drug, date of therapy initiation, and type of drug administration.

This study was performed following the guidelines of the latest version of the Declaration of Helsinki, with the study protocol accepted by the Ethics Committee of the University Hospital of Split on 5 April 2023 (Ur.no. 500-03/23-01/84).

### 2.2. Inclusion and Exclusion Criteria

The inclusion criteria were as follows:Patients with biologic DMARDs prescribed in the period from 1 July 2018 to 31 March 2020.Patients with biologic DMARDs prescribed in the period from 1 April 2020 to 21 December 2021.Patients older than 18 years.

Therefore, the exclusion criteria were as follows:Patients with no biologic DMARDs prescribed in the period from 1 July 2018 to 31 March 2020.Patients with no biologic DMARDs prescribed in the period from 1 April 2020 to 21 December 2021.Patients younger than 18 years.

### 2.3. Statistical Analysis

Data were presented as numbers and percentages. To test for differences between groups (pre-COVID-19 vs. COVID-19), the chi-square test was used. For situations with fewer than 10 cases per group, Fisher’s exact test was used. A *p*-value of less than 0.05 was considered statistically significant. All statistical analysis was performed using the free software environment for statistical computing R version 4.0.0 [17].

## 3. Results

After applying the inclusion and exclusion criteria, the Outpatient Clinic for Rheumatology at the University Hospital of Split, Croatia, treated 209 patients prescribed biological DMARDs in the period from 1 July 2018 to 31 March 2020 (pre-COVID-19), with an average age of 50.67 ± 13.95 years, of whom 146 were female (70%). In the period from 1 April 2020 to 21 December 2021 (COVID-19), 185 patients who were prescribed biologic DMARDs for various rheumatologic conditions were treated (11.5% fewer patients than in the pre-COVID-19 period), with an average age of 51.53 ± 13.67 years, of whom 124 were female (67%). In the pre-COVID-19 period, 231 biologic DMARDs were prescribed, while 204 were prescribed in the COVID-19period, including etanercept, infliximab, adalimumab, certolizumab, golimumab, tocilizumab, secukinumab, ixekizumab, and sarilumab, while ustekinumab was prescribed only in the period before COVID-19.

The prescriptions of each biologic DMARD for each patient are summarised in Table 1. As shown, only three drugs reached statistical significance, i.e., infliximab and ixekizumab were prescribed more frequently (*p* = 0.048 and *p* < 0.001, respectively), while tocilizumab was prescribed less frequently, in the period of COVID-19 (*p* < 0.001).

In addition, the drugs were grouped according to their mechanism of action, and the prescription trends in each group were analysed. Etanercept, infliximab, adalimumab, certolizumab, and golimumab were grouped and analysed as TNF-α inhibitors; tocilizumab and sarilumab were grouped and analysed as IL-6 inhibitors; and finally, secukinumab and ixekizumab were grouped and analysed as IL-17A inhibitors. There was no statistical significance in the prescribing of TNF-α inhibitors between the period prior to COVID-19 and during COVID-19 (142 (61%) vs. 122 (60%) prescriptions, *p* = 0.722). IL-6 inhibitors were statistically less prescribed in the COVID-19 period (48 (21%) vs. 21 (10%) prescriptions, *p* = 0.003), while IL-17A inhibitors were statistically more prescribed in the COVID-19 period (39 (17%) vs. 61 (30%) prescriptions, *p* = 0.001).

Biosimilar prescribing was analysed by the brand name of the drug prescribed. In the period prior to COVID-19, a total of 64 (28%) biosimilars were prescribed, while in the COVID-19period, a total of 81 (40%) biosimilars were prescribed, showing an increase in biosimilar prescriptions with statistical significance (*p* = 0.008). Table 2 shows a list of original drugs and biosimilars approved by the European Medicines Agency (EMA) for use in Croatia, as well as a summary of their use in the pre-COVID-19 and COVID-19 periods, according to our data [18].

Regarding the route of drug administration, in the period before COVID-19, 48 (21%) prescriptions were administered intravenously and 183 (79%) prescriptions were administered subcutaneously. In the COVID-19 period, 16 (8%) prescriptions were administered intravenously and 188 (92%) prescriptions were administered subcutaneously. As shown, there is a statistically significant increase in prescriptions administered subcutaneously, while intravenous administration decreased during the COVID-19 period (*p* < 0.001).

Figure 1 shows a bar plot summarizing all individual drugs prescribed, mechanisms of action, and routes of administration for the pre-COVID-19 and COVID-19 periods.

The diagnoses for which medications were prescribed included ICD-10 codes M05, M05.1, M05.3, M05.7, M05.8, M05.9, M06, M06.1, M06.9, M07, M07.1, M07.2, M07.3, M08, M08.9, M09, M13, M31, M31.6, M45, M46, M46.1, M46.9, M47, and M49. M05, M05.1, M05.3, M05.7, M05.8, M05.9, M06, and M06.9 were combined for analysis (rheumatoid arthritis (RA)); M07, M07.1, M07.2, and M07.3 were combined for analysis (psoriatic arthritis (PsA)); M08, M08.9, and M09 were combined for analysis (juvenile arthritis (JA)); M47 and M45 were combined for analysis (ankylosing spondylitis (AS)); M46, M46.1, M46.9, and M49 were combined for analysis (spondylopathies); and M31 and M31.6 were combined for analysis (necrotizing vasculopathies); M06.1 (adult-onset Still’s disease) and M13 (other arthritis) were analyzed separately.

The pre-COVID-19 period included 91 (43.5%) patients with RA, 35 (17%) patients with PsA, 8 (4%) patients with JA, 46 (22%) patients with AS, 23 (11%) patients with other spondylopathies, 2 (1%) patients with necrotizing vasculopathies, 1 (0.5%) patient with adult-onset Still’s disease, and 3 (1%) patients with other arthritis. The COVID-19 period included 56 (30%) patients with RA, 59 (32%) patients with PsA, 2 (1%) patients with JA, 52 (28%) patients with AS, 13 (7%) patients with other spondylopathies, 1 (1%) patient with necrotizing vasculopathies, and 2 (1%) patients with adult-onset Still’s disease.

The indications for treatment with specific biologic DMARDs were as follows: etanercept was approved for the treatment of RA, JA, PsA, AS, and psoriasis; infliximab for the treatment of PsA, AS, RA, and psoriasis; adalimumab for the treatment of RA, AS, JA, PsA, and psoriasis; certolizumab for the treatment of RA, AS, PsA and psoriasis; golimumab for the treatment of RA, PsA, and AS; tocilizumab for the treatment of RA; secukinumab for the treatment of PsA, AS, JA and psoriasis; and sarilumab for the treatment of RA, but only in combination with methotrexate in both the pre-COVID-19 and COVID-19 periods. Ixekizumab was approved for the treatment of psoriasis and PsA in the period before COVID-19, while it was also approved for the treatment of AS in the COVID-19 period. Ustekinumab was only approved for the treatment of Crohn’s disease and ulcerative colitis in both periods and was only used to treat the underlying disease in enteropathic arthropathy in the pre-COVID-19 period [18].

When specific drug prescriptions were analyzed by diagnosis, the only differences that reached statistical significance were found in AS and RA. In AS, more adalimumab was prescribed in the pre-COVID-19 period (25 vs. 15 patients, *p* = 0.010), while ixekizumab was prescribed less (1 vs. 10 patients, *p* = 0.009). In RA, even though not reaching statistical significance, overall IL-6 inhibitors were prescribed less during COVID-19 (40 vs. 16 patients, *p* = 0.062). However, specifically for RA, tocilizumab was prescribed less in the period of COVID-19 (34 vs. 10 patients, *p* = 0.012).

## 4. Discussion

As the results of this study show, the number of patients prescribed biologic DMARDs has decreased overall, and consequently, fewer biologic DMARDs were prescribed during COVID-19 than before COVID-19, due to the reduction in hospital outpatient hours and the reduced availability of physicians working in COVID-19 wards [5]. Regarding specific treatment with biologic DMARDs, IL-6 receptor inhibitors were prescribed less frequently (lower prescription of tocilizumab), whereas IL-17A inhibitors were prescribed more frequently during COVID-19 (higher prescription of ixekizumab). Overall, the data suggest a shift from tocilizumab to ixekizumab, but this may not be the case, as these drugs have completely different indications. Therefore, shifts in prescribing should be interpreted considering other factors such as patient structure, availability, route of administration, indications, and shifts within a particular rheumatologic condition, which will be discussed.

The IL-6 receptor inhibitor tocilizumab is an effective drug for the treatment of severe, active, and progressive RA in patients not previously treated with methotrexate [19]. However, IL-6 receptor inhibitors have also demonstrated their potential value in the treatment of COVID-19: their administration was associated with a reduced risk of death, particularly in severe conditions [20,21]. As tocilizumab was used as COVID-19 treatment, this led to shortages reported in European countries for the treatment of rheumatic diseases [22]. As the results of this study show, the decrease in the prescribing of IL-6 inhibitors in the treatment of RA was mainly due to a decrease in tocilizumab prescription for RA, supporting these claims. In addition, the number of RA patients treated with biologic DMARDs was found to decrease overall during COVID-19, which may also be due to the decreased availability of tocilizumab, as this drug is only approved for the treatment of RA. Overall, prescribing trends should be interpreted in light of the decrease or increase in the indications for which they are being treated, which have been analyzed and will be discussed further.

IL-17A inhibitors, on the other hand, play a central role in the treatment of PsA and AS [8]. As the results show, the number and percentage of PsA patients treated increased in the period of COVID-19, and so did the prescription of IL-17A inhibitors, ixekizumab and others, as the therapy of choice in PsA. Epidemiologic trends of PsA in recent years vary worldwide, with some studies reporting higher incidence rates, while most studies report consistent incidence rates in the years studied. What is certain, however, is that the prevalence of PsA is steadily increasing, even if the overall incidence rate does not vary [23,24,25]. While it is possible that there is indeed a true increase in disease expression, all that is reported is most likely due to underdiagnosis, as there is no golden standard for PsA diagnosis, making diagnosis difficult due to heterogeneous expression and misclassification in the use of diagnostic codes, as these patients also have other reasons for developing joint pain (osteoarthritis, fibromyalgia, and gout) [26]. In our institution specifically, even though the majority of rheumatologists and dermatologists were assigned to work in COVID units, the collaboration between these two specialties has improved thanks to the organization established right before the COVID-19pandemic. Owing to the application of questionnaires used in the early detection of PsA, dermatologists actually referred more patients to rheumatologists, and therefore it is only logical that the number of PsA patients increased as more PsA was detected.

During the period of COVID-19, an increase in infliximab prescription was also noted. As patients were mostly managed and consulted in the virtual outpatient clinic, they often expressed concern about not having face-to-face contact, and in some cases, preferred the choice of medication with intravenous administration. Infliximab formulations, original and biosimilar drugs, are administered intravenously only, and as shown previously, the only intravenous formulations available were tocilizumab and infliximab. Because there were shortages of tocilizumab and it was predominantly used to treat COVID-19, there has been an increase in the use of infliximab. When examining patient preference for RA treatment in Japan via an online survey, it was found that more than half of participants wanted to change their treatment method when their RA symptoms changed, with a higher percentage of those using self-administered subcutaneous injections. In addition, a higher percentage of patients using self-administered subcutaneous injections, compared with those who used other forms of treatment administration, said they would prefer to change their method of administration as they get older. All of this supports patients’ general fear of continuing self-administered subcutaneous injections as they age, as older people prefer their treatment to be administered by healthcare providers [12,27].

However, although patients preferred face-to-face counseling, subcutaneous use inevitably increased because of complications associated with intravenous use and limited outpatient clinic hours and physician availability [5]. These findings are consistent with trends in other parts of the world: in the period of COVID-19, the National Institute for Health and Care Excellence in the United Kingdom published new guidelines recommending that patients receiving intravenous treatment be assessed for a switch to the same treatment in subcutaneous form. That provided many benefits: a cost analysis showed that the cost of subcutaneous administration was 50% lower than infusion, waiting times in the infusion ward were shorter, and the risk of COVID-19 infection was lower. However, the switch to subcutaneous administration increased pressure on home care and primary care services, highlighting the need for a well-structured primary care system [28]. Regarding the comparison of treatment effectiveness in relation to route of administration, studies are sparse and very specific to particular drugs in only one or two indications, with a lack of systematic reviews and meta-analyses. There are no studies investigating a switch of therapeutic agent (generic) to facilitate the route of administration, i.e., switching from tocilizumab to etanercept. However, switching from intravenous to subcutaneous use of tocilizumab was found to be safe, effective, and well tolerated in one study [29]. Another study in which tocilizumab was administered intravenously and subcutaneously showed similar improvements in RA symptoms and quality of life [30]. Further study of systemic lupus erythematosus treatment and belimumab use found that switching from intravenous to subcutaneous use was successful in all cases and had no effect on quality of life or disease activity [31]. All of these results suggest that, overall, the routes of administration are comparable as treatment options in rheumatologic conditions, although we cannot state this with absolute certainty. The specific impact of this change on other formulations and overall efficacy requires further investigation, as the data are inconclusive.

Overall, as the use of subcutaneous application increased, the use of biosimilars also increased in the COVID-19 period. The use of biosimilars is beneficial because biosimilars are less expensive than the original compound, and in several studies, they proved to have the same effect on patient quality of life and symptom alleviation [32,33]. Biosimilars were also widely available in the COVID-19 period, as they were associated with cost reductions and thus proved invaluable for the treatment of rheumatologic conditions [34].

In the treatment of AS, there was a significant increase in ixekizumab prescription, while at the same time, there was a decrease of adalimumab administration, in the COVID-19 period. TNF-α inhibitors are a preferred treatment for AS and, according to recent meta-analysis by Yu et al., drug survival is comparable among all TNF-α inhibitors used to treat AS [35]. However, ixekizumab emerged as a new therapeutic choice approved for use in AS, which, together with secukinumab, provided a new route of treatment besides TNF-α inhibitors [18]. In one clinical trial, ixekizumab proved to achieve similar reduction in disease activity as well as symptomatology [36]. This provided a therapy option for patients that had a primary failure, secondary failure, or intolerance of TNF-α inhibitors.

One of the limitations of this study is the cross-sectional design, through which only association, but not causality, can be established. In addition, data on COVID-19 infection history, vaccinations, comorbidities, level of patient satisfaction, clinical status, and other medications were not collected, which may also help in understanding overall decision making and prescribing affinity. Although COVID-19 has affected the availability of health care, there has been no significant change in the prescribing practices of biologic DMARDs for rheumatologic diseases at the University Hospital of Split, which means that the standard of care has been maintained, although the conditions were unprecedented and difficult. As reported in another study on patient satisfaction and effectiveness of virtual visits compared to face-to-face visits, patient and physician engagement was comparable, and patients are open to the higher level of technology use [12]. Furthermore, the future role of telemedicine in standard patient care is highly dependent on patient preferences, so an individualistic approach and identification of patient preferences play an important role in its application [37]. Currently, however, telemedicine and virtual visits are proving to be valuable tools for maintaining a high level of quality care under extreme and unprecedented conditions. Although this study did not find major differences in the use of medications during the pandemic, this study points to the importance of further studies to evaluate the impact of switching and using biologic DMARDs during COVID-19 and their potential impact on future treatment.

## Figures and Tables

**Figure 1 jpm-13-01199-f001:**
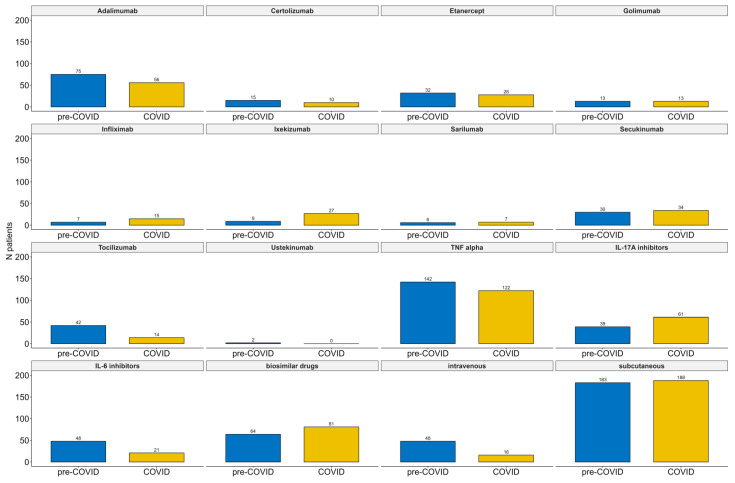
Summary of individual medication analysis, mechanisms of action, and route of administration regarding pre-COVID-19 and COVID-19 periods.

**Table 1 jpm-13-01199-t001:** Summary of patient biologic DMARD prescription in pre-COVID-19 and COVID-19 periods.

Biologic DMARDs	Pre-COVID-19 * (*n* = 209)	COVID-19 * (*n* = 185)	*p*-Value
Etanercept	32 (15)	28 (15)	0.961
Infliximab	7 (3)	15 (8)	0.048
Adalimumab	75 (36)	56 (30)	0.238
Certolizumab	15 (7)	10 (5)	0.472
Golimumab	13 (6)	13 (7)	0.747
Tocilizumab	42 (20)	14 (8)	<0.001
Secukinumab	30 (14)	34 (18)	0.280
Ixekizumab	9 (4)	27 (15)	<0.001
Sarilumab	6 (3)	7 (4)	0.779
Ustekinumab	2 (1)	0 (0)	/

* data showed as number (%) of patients prescribed.

**Table 2 jpm-13-01199-t002:** Summary of biosimilars and their original drugs used in the pre-COVID-19 and COVID-19 periods.

Biologic DMARDs *	Brand Name of the Drug	Pre-COVID-19	COVID-19	Application	Biosimilar or Original
Etanercept	Erelzi	Approved/not used	Approved/used	Subcutaneous	Biosimilar
	Benepali	Approved/used	Approved/used	Subcutaneous	Biosimilar
	Enbrel	Approved/used	Approved/used	Subcutaneous	Original
	Nepexto	Not approved	Approved/used	Subcutaneous	Biosimilar
Infliximab	Flixabi	Approved/used	Approved/used	Intravenous	Biosimilar
	Remicade	Approved/used	Approved/used	Intravenous	Original
	Remsima	Approved/used	Approved/used	Intravenous **	Biosimilar
	Inflectra	Approved/used	Approved/used	Intravenous	Biosimilar
	Zessly	Approved/not used	Approved/used	Intravenous	Biosimilar
Adalimumab ***	Amgevita	Approved/used	Approved/used	Subcutaneous	Biosimilar
	Hulio	Approved/used	Approved/used	Subcutaneous	Biosimilar
	Humira	Approved/used	Approved/used	Subcutaneous	Original
	Imraldi	Approved/used	Approved/used	Subcutaneous	Biosimilar
	Hyrimoz	Approved/used	Approved/used	Subcutaneous	Biosimilar
	Idacio	Approved/used	Approved/used	Subcutaneous	Biosimilar

* Etanercept, infliximab, and adalimumab were the only drugs used to treat rheumatologic conditions that had biosimilar drugs approved for use in Croatia; other drugs are not depicted in this table. ** Remsima formulation for subcutaneous use was not available even though approved. *** Hefiya, Hukyndra, Yuflyma, Amsparity, and Libmyris formulations were not used regardless of the EMA approval.

## Data Availability

All data is available upon reasonable request to the corresponding author.

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
