# Peer review of "Prescription Trends of Biologic DMARDs in Treating Rheumatologic Diseases: Changes of Medication Availability in COVID-19"

_jpm, 2023, doi:10.3390/jpm13081199_

Round 1

Reviewer 1 Report

The authors have studied the effects of the COVID pandemic on bDMARD use in Croatia. They looked at bDMARD use before and immediately after the onset of the COVID pandemic. The authors collected data on bDMARD prescribing for a 20 month period pre-pandemic and a 20 month period immediately post the onset of the pandemic

The authors describe the challenging situation at the start of the pandemic – with redeployment of staff, re-organisation of in-patient facilities and the move to virtual out-patient consultation.

They found a non-significant 10.5% reduction in the number of patients prescribed bDMARDs but a change in the type of biologic used – with a reduction in the use of IL-6 inhibitors (54% reduction) and a 60% increase in the use of IL-17A inhibitors. There was a significant move away from IV preparations in favour of s/c preparations overall but no significant change in the use of TNF inhibitors. The reduction in IL-6 inhibitor use may be partly explained by the use of IL6-inhibitors as treatment for severe COVID-19 infection resulting in a relative shortage of the drug across Europe.

Author Response

Response to the Reviewer 1

The authors have studied the effects of the COVID pandemic on bDMARD use in Croatia. They looked at bDMARD use before and immediately after the onset of the COVID pandemic. The authors collected data on bDMARD prescribing for a 20 month period pre-pandemic and a 20 month period immediately post the onset of the pandemic

The authors describe the challenging situation at the start of the pandemic – with redeployment of staff, re-organisation of in-patient facilities and the move to virtual out-patient consultation.

They found a non-significant 10.5% reduction in the number of patients prescribed bDMARDs but a change in the type of biologic used – with a reduction in the use of IL-6 inhibitors (54% reduction) and a 60% increase in the use of IL-17A inhibitors. There was a significant move away from IV preparations in favour of s/c preparations overall but no significant change in the use of TNF inhibitors. The reduction in IL-6 inhibitor use may be partly explained by the use of IL6-inhibitors as treatment for severe COVID-19 infection resulting in a relative shortage of the drug across Europe.

Dear Reviewer 1,

Thank you for your kind comments and for taking the time to evaluate our work. We revised and updated our study according to all reviewers comments and we hope we further increased the quality of our research.

Reviewer 2 Report

Dear Authors,

You describe the challenges of the COVID-19 pandemic for the care of your rheumatology patients with, among other things, video consultations and reduced contacts and, as one indicator, you have performed a retrospective analysis of prescriptions of biologics in 21 months before and 21 months after the onset of the COVID-19 pandemic in Croatia. Only patients who were prescribed biologics at all were evaluated.

There was a slight decrease in the number of patients served, and 2 major trends in the rate of prescriptions; tocilizumab prescriptions decreased significantly, and ixekizumab prescriptions increased.

You interpret the decrease in tocilizumab as competition from prescriptions for affected patients with COVID-19. The data suggest a shift from tocilizumab to ixekizumab, but this cannot exist because of entirely different indications.

The study has some major flaws that significantly limit the power.

1.) In addition to the COVID-19 pandemic, there were new drug developments whose use would have increased even without the pandemic (e.g., biosimilars) and, in particular, a growing treatment dogma for psoriatic arthritis. This may explain, independently of COVID-19, an increase in the number of patients with psoriatic arthritis and the substances approved for this purpose, also described by the authors.

2.) There are not meaningful groupings: A possible assignment of CED associated arthritides (M07) to psoriatic arthritis is questionably reasonable, and an assignment of AOSD to RA is not reasonable. Finally, due to the different group composition before and under COVID-19, a global comparison of the individual substances cannot be meaningful.

3.) There are numerous patients included who received biologics outside the tested and approved indication (e.g. M35, M47, M54, D86). 

4.) The number of RA patients treated with biologics has decreased significantly by 1/3. This is not expected with a stable need for care. There is no explanation or comment for this. Since tocilizumab is (only) approved for RA, this alone can explain a reduction. 

5.) There is no comparison to the total number of patients treated - i.e. those without therapy or those treated with conventional DMARDs.

In addition, there are reports that I cannot understand: The number of i.v. applications decreased in the study, but infliximab, the only drug to be applied i.v., increased.

In figure 1 the COVID-row is missing for Ustekinumab (a zero-row)

In summary, the significance of the data is very limited and not very helpful for the reader. Compared to the essence of the data, the text is very long.

The grammar should be checked by a native speaker.

Author Response

Response to the Reviewer 2

Dear Reviewer 2,

Thank you for your kind comments and for taking the time to evaluate our work. We revised and updated the manuscript following your instructons in detail, we hope to your satisfaction.

1) Dear Authors,

You describe the challenges of the COVID-19 pandemic for the care of your rheumatology patients with, among other things, video consultations and reduced contacts and, as one indicator, you have performed a retrospective analysis of prescriptions of biologics in 21 months before and 21 months after the onset of the COVID-19 pandemic in Croatia. Only patients who were prescribed biologics at all were evaluated.

There was a slight decrease in the number of patients served, and 2 major trends in the rate of prescriptions; tocilizumab prescriptions decreased significantly, and ixekizumab prescriptions increased.

You interpret the decrease in tocilizumab as competition from prescriptions for affected patients with COVID-19. The data suggest a shift from tocilizumab to ixekizumab, but this cannot exist because of entirely different indications.

The trends in prescription as well as the profile of treated patients changed and therefore we rephrased the part of the discussion that now states:

„The data overall suggests a shift from tocilizumab to ixekizumab, but that could not be the case as these drugs have entirely different indications. Therefore, shifts in prescription should be interpreted taking in account other factors such as patient structure, availability, route of administration, indications, and shifts within a specific rheumatologic condition.“

2) In addition to the COVID-19 pandemic, there were new drug developments whose use would have increased even without the pandemic (e.g., biosimilars) and, in particular, a growing treatment dogma for psoriatic arthritis. This may explain, independently of COVID-19, an increase in the number of patients with psoriatic arthritis and the substances approved for this purpose, also described by the authors.

To ascertain exactly which use of biosimilars increased and if development and approval of new drugs had any effect at all we added Table 2. which states:

Table 2. Summary of biosimilars and their original drugs used in pre-COVID-19 and COVID-19 period.

Biologic DMARDs*

Brand name of the drug

Pre-COVID-19

COVID-19

Application

Biosimilar or original

Etanercept

Erelzi

Approved/not used

Approved/used

Subcutaneous

Biosimilar

Benepali

Approved/used

Approved/used

Subcutaneous

Biosimilar

Enbrel

Approved/used

Approved/used

Subcutaneous

Original

Nepexto

Not approved

Approved/used

Subcutaneous

Biosimilar

Infliximab

Flixabi

Approved/used

Approved/used

Intravenous

Biosimilar

Remicade

Approved/used

Approved/used

Intravenous

Original

Remsima

Approved/used

Approved/used

Intravenous**

Biosimilar

Inflectra

Approved/used

Approved/used

Intravenous

Biosimilar

Zessly

Approved/not used

Approved/used

Intravenous

Biosimilar

Adalimumab***

Amgevita

Approved/used

Approved/used

Subcutaneous

Biosimilar

Hulio

Approved/used

Approved/used

Subcutaneous

Biosimilar

Humira

Approved/used

Approved/used

Subcutaneous

Original

Imraldi

Approved/used

Approved/used

Subcutaneous

Biosimilar

Hyrimoz

Approved/used

Approved/used

Subcutaneous

Biosimilar

Idacio

Approved/used

Approved/used

Subcutaneous

Biosimilar

*Etanercept, infliximab and adalimumab were the only drugs used to treat rheumatologic diseases that had biosimilar drugs approved for use in Croatia.

**Remsima formulation for subcutaneous use was not available even though approved.

***Hefiya, Hukyndra, Yuflyma, Amsparity and Libmyris formulations were not used regardless of the EMA approval.

To further discern and compare the treatment dogma for psoriatic arthritis and analyse the changes in number of treated patients we added the following paragraph:

“IL-17A inhibitors, on the other hand, are central in treating PsA and AS [8]. As showed in the results, there is an increase in numbers and in percentage of PsA patients treated in COVID-19 period and therefore, an increased prescription of IL-17A inhibitors, ixekizumab and others, as a therapeutic choice for PsA. The epidemiology trends of PsA in recent years vary globally, some studies report higher incidence rates while most studies report steady incidence rates through the years studied. However, it is certain that even though overall incidence rate did not vary, the prevalence of PsA increases steadily [21, 22, 23]. While it is possible that there is a true increase in disease expression, all reported is most likely due to underdiagnosis as there is no golden standard for PsA diagnosis which makes it challenging due to its heterogenous manifestation, and misclassification when using diagnostic codes as these patients have also other reasons for developing joint pain (osteoarthritis, fibromyalgia, and gout) [24]. In our institution specifically, even though the majority of rheumatologists and dermatologists were assigned to work in COVID units, the collaboration between these two specialties improved due to organization assignments set up right before the COVID pandemic. Thanks to the application of questionnaires used in early screening for PsA, dermatologists indeed refered more patients to rheumatologists and therefore it is only logical that the number of PsA patients increased as more PsA was detected.”

3) There are not meaningful groupings: A possible assignment of CED associated arthritides (M07) to psoriatic arthritis is questionably reasonable, and an assignment of AOSD to RA is not reasonable. Finally, due to the different group composition before and under COVID-19, a global comparison of the individual substances cannot be meaningful.

There are numerous patients included who received biologics outside the tested and approved indication (e.g. M35, M47, M54, D86).

Thank you for your observation. We analyzed and regrouped the analysis accordingly – M06.1 was analysed separately while in reviewing D86 we came to the conclusion that this patient had rheumatoid arthritis but was protocoled in the paperwork under this code by mistake. Furthermore, code M47 was joined in analysis with ankylosing spondylitis as that was the underlying diagnosis. The two patients with diagnoses under M35 and M54 were excluded as by further examination it was seen they recieved biologics for non-rheumatologic indications. We further analysed the M07 group and came to the conclusion that in fact only 2 patients in pre-COVID period (0 patients in COVID period) had enteropathic arthropathy and were treated with ustekinumab for their underlying disease while all others were definitely in psoriatic arthritis group so the analysis was left as is as they were mostly only patients with psoriatic arthritis. The results were corrected accordingly. Finally, the global comparison is merely an indication of the direction of said changes and to better comprehend these changes we analysed patient and drug prescription dynamic. To further elucidate these differences we added a paragraph in the discussion section which now states:

„Regarding specific treatment with biologic DMARDs, IL-6 receptor inhibitors were less prescribed (lower prescription of tocilizumab) while IL-17A inhibitors were more fre-quently prescribed (higher prescription of ixekizumab) in COVID-19 period. The data overall suggests a shift from tocilizumab to ixekizumab, but that could not be the case as these drugs have entirely different indications, therefore, shifts in prescription should be interpreted taking in account other factors such as patient structure, availability, route of administration, indications, and shifts within a specific rheumatologic condition which will be discussed.

IL-6 receptor inhibitor tocilizumab is an effective drug for severe, active, and progres-sive RA in patients not previously treated with methotrexate [17]. However, the IL-6 re-ceptor inhibitors also proved their potential value in treating cases of COVID-19 – their administration was associated with a reduced risk of mortality, especially in severe con-ditions [18, 19]. Therefore, as tocilizumab was used as a COVID-19 treatment it led to shortages reported in European countries for use in rheumatic disease treatment [20]. As shown in the results of this study, a reduction in prescription of IL-6 inhibitors in treating RA was mainly due to reduced prescription of tocilizumab for RA support these claims. Furthermore, it was noted a decrease in number of treated RA patients with bDMARDs in COVID-19 period overall which can also be due to decreased availability of tocilizumab as this drug is only approved for treating RA. Overall, the prescription trends should be in-terpreted taking in account decrease or increase of indications for which they are treated which were analyzed and are further discussed.“

4) The number of RA patients treated with biologics has decreased significantly by 1/3. This is not expected with a stable need for care. There is no explanation or comment for this. Since tocilizumab is (only) approved for RA, this alone can explain a reduction.

Thank you for your observation. To further elucidate these differences we added a sentence in the discussion section which states:

„Furthermore, it was noted a decrease in number of treated RA patients with bDMARDs in COVID-19 period overall which can also be due to decreased availability of tocilizumab as this drug is only approved for treating RA.“

5) There is no comparison to the total number of patients treated - i.e. those without therapy or those treated with conventional DMARDs.

Unfortunately, the large database where we pooled all data, accounts only for patients which recieved biologic treatment. The data about those treated with non-biologics was unavailable in this database.

6) In addition, there are reports that I cannot understand: The number of i.v. applications decreased in the study, but infliximab, the only drug to be applied i.v., increased.

The increase happened due to some patient anxiety in which they insisted to be treated in a hospital enviroment. Detailed explanation in discussion states:

„Increased prescription of infliximab was also noticed in COVID-19 period. As pa-tients were mostly followed up and consulted in the virtual outpatient clinic, frequently patients expressed the anxiety of no face-to-face contact and in some cases preferred the choice of medication with intravenous application. Infliximab formulations, original and biosimilars, are only administered intravenously, and, as previously shown, the only formulations administered intravenously available were tocilizumab and infliximab. As there were shortages of tocilizumab and it was predominantly used for treating COVID-19, there is an increase in infliximab use. When investigating patient preference for RA treatment in Japan by online survey, they found that more than one half of the par-ticipants wanted to change their treatment mode if their RA symptoms changed, with a higher rate among those using self-administered subcutaneous injections. Additionally, a higher percentage of patients using self-administered subcutaneous injection compared to those who used other modes of treatment administration indicated that they would prefer a change in their method of administration as they get older. All of this supports the over-all patients anxiety about continuing self-administered subcutaneus injection as they age with elderly people preferring their treatment to be administered by healthcare providers [11, 25].“

7) In figure 1 the COVID-row is missing for Ustekinumab (a zero-row)

Figure 1. was updated accordingly.

8) In summary, the significance of the data is very limited and not very helpful for the reader. Compared to the essence of the data, the text is very long.

We hope that in our corrections we managed to add value to our research. The text longevity is a requirement of the journal as they have a limitation of 4000 words minimum per article.

Reviewer 3 Report

The authors presented a cross-sectional study on the use of biological DMARDs during the Covid 19 epidemic. Unfortunately, they did not provide any clinical data, apart from ICD 10 codes, regarding therapeutic decisions. It is known that decisions to initiate, change or withdraw treatment are based on diagnosis, clinical presentation, efficacy/loss of efficacy, availability and side effects. In this approach, the presented study does not make sense, it is impossible to draw specific conclusions regarding the described changes. The authors themselves noticed their limitation – only an association can be established but not causality. In my opinion, the text is not suitable for publication.

Author Response

Response to the Reviewer 3

Reviewer comments: The authors presented a cross-sectional study on the use of biological DMARDs during the Covid 19 epidemic. Unfortunately, they did not provide any clinical data, apart from ICD 10 codes, regarding therapeutic decisions. It is known that decisions to initiate, change or withdraw treatment are based on diagnosis, clinical presentation, efficacy/loss of efficacy, availability and side effects. In this approach, the presented study does not make sense, it is impossible to draw specific conclusions regarding the described changes. The authors themselves noticed their limitation – only an association can be established but not causality. In my opinion, the text is not suitable for publication.

Dear Reviewer 3,

Thank you for taking the time to review our work. We made some significant changes and added as much information about the trends and availability of biologic drugs used in treatment of rheumatologic diseases. We hope that the revised version would be more suitable for publication.

Reviewer 4 Report

The authors described the prescription patterns of biologic DMARDs before and after the official start of COVID pandemic.

There are many points to improve:

-      -    the prescription rate for many drugs is not strictly dependent by COVID era (e.g. the indication of ixekizumab for ankylosing spondylitis is post COVID). In case of therapeutic change, the authors should indicate the motivation

-        -  the preference of the subcutaneous formulation over the intravenous one has been considered only for drugs with dual formulation available? (what about subcutaneous infliximab?)

-         - the availability of drugs before and after the start of COVID, and their approval for different indications

-        -  the evaluation of the clinical response to the various prescribed drugs is lacking

Author Response

Response to the Reviewer 4

Dear Reviewer 4,

Thank you for your kind comments and for taking the time in reviewing our manuscript. We revised and updated the manuscript following your comments.

1) The authors described the prescription patterns of biologic DMARDs before and after the official start of COVID pandemic.

There are many points to improve:

- the prescription rate for many drugs is not strictly dependent by COVID era (e.g. the indication of ixekizumab for ankylosing spondylitis is post COVID). In case of therapeutic change, the authors should indicate the motivation

Thank you for your comment. As we were unable to include specific clinical data for each patient we included all trends in changes of indications and treatment choices. We also included the limitations section that states:

„One of the limitations of this study is the cross-sectional design by which only an as-sociation can be established but not causality. Also, the data about COVID-19 infection history, vaccination, comorbidities, level of patient satisfaction, clinical status, and other medication was not collected which can be also a contributing factor in comprehending overall decision making and prescription affinity. Overall, even though COVID-19 affected the availability of health care, the prescription trends of biologic DMARDs for rheumato-logic diseases did not vary significantly in University Hospital of Split, meaning the standard of care was maintained even though conditions were unprecedented and diffi-cult.“

2)  the preference of the subcutaneous formulation over the intravenous one has been considered only for drugs with dual formulation available? (what about subcutaneous infliximab?)

As seen in added Table 2., the Remsima formulation for subcutaneous use, approved right around the COVID start, was not available for use in COVID period in our institution. The trends in prescription of subcutaneous and intravenous formulation were therefore observed in general as only tocilizumab and infliximab were available for intravenous use while the rest of the medications were subcutaneously applied, it was not possible to analyze any drug specifically.

3) the availability of drugs before and after the start of COVID, and their approval for different indications

Thank you for your observation. To better discern these differences we added the Table 2. which states:

Table 2. Summary of biosimilars and their original drugs used in pre-COVID-19 and COVID-19 period.

Biologic DMARDs*

Brand name of the drug

Pre-COVID-19

COVID-19

Application

Biosimilar or original

Etanercept

Erelzi

Approved/not used

Approved/used

Subcutaneous

Biosimilar

Benepali

Approved/used

Approved/used

Subcutaneous

Biosimilar

Enbrel

Approved/used

Approved/used

Subcutaneous

Original

Nepexto

Not approved

Approved/used

Subcutaneous

Biosimilar

Infliximab

Flixabi

Approved/used

Approved/used

Intravenous

Biosimilar

Remicade

Approved/used

Approved/used

Intravenous

Original

Remsima

Approved/used

Approved/used

Intravenous**

Biosimilar

Inflectra

Approved/used

Approved/used

Intravenous

Biosimilar

Zessly

Approved/not used

Approved/used

Intravenous

Biosimilar

Adalimumab***

Amgevita

Approved/used

Approved/used

Subcutaneous

Biosimilar

Hulio

Approved/used

Approved/used

Subcutaneous

Biosimilar

Humira

Approved/used

Approved/used

Subcutaneous

Original

Imraldi

Approved/used

Approved/used

Subcutaneous

Biosimilar

Hyrimoz

Approved/used

Approved/used

Subcutaneous

Biosimilar

Idacio

Approved/used

Approved/used

Subcutaneous

Biosimilar

*Etanercept, infliximab and adalimumab were the only drugs used to treat rheumatologic diseases that had biosimilar drugs approved for use in Croatia.

**Remsima formulation for subcutaneous use was not available even though approved.

***Hefiya, Hukyndra, Yuflyma, Amsparity and Libmyris formulations were not used regardless of the EMA approval.

To further analyze indications for which these drugs were approved we added the following paragraph:

„The indications for treatment with specific bDMARDs were as follows: etanercept was approved for use in RA, JA, PsA, AS and psoriasis, infliximab was approved for use in PsA, AS, RA and psoriasis, adalimumab was approved for use in RA, AS, JA, PsA and psoriasis, certolizumab was approved for use in RA, AS, PsA and psoriasis, golimumab was approved for use in RA, PsA and AS, tocilizumab was approved for use in RA, secukinumab was approved for use in PsA, AS, JA and psoriasis, and sarilumab was ap-proved for use in RA but only in combination with methotrexate in both pre-COVID-19 and COVID-19 period. Ixekizumab was in pre-COVID-19 period approved for use in pso-riasis and PsA while in COVID-19 period it was also approved for use in AS. Usteki-numab was in both periods only approved for use in Chron’s disease and ulcerative colitis and was used only to treat the underlying disease in case of enteropathic arthropathy in pre-COVID-19 period.“

4)  the evaluation of the clinical response to the various prescribed drugs is lacking

All prescribed drugs and the dynamic of treatment according to clinical response was done following the EULAR recommendations. However, we were unable to pool the specifics regarding the patients in our study as it was not recorded in this specific database that contained data about biologic treatment, that data was recorded in medical documentation of these patients.

Round 2

Reviewer 2 Report

Dear Authors,

thank you very much for the revision of the manuscript. Some inaccuracies regarding diagnoses and classifications could be reduced, but the relevant limitations remain. unfortunately, the linguistic quality was not improved in a way that would allow barrier-free reading. A native-language revision could increase the informative value.

I cannot understand some of the changes: On page 3, line 147, the percentage of women was shortened from 69.85 to 67%, although with 146 women and 209 patients the percentage remained the same, correct would be 70%. The percentage of TNF prescriptions (page 4 line 169) was increased from 61.47 to 62%, rounded 61% would be correct. In line 172, the proportion of IL17 prescriptions was increased from 16.47 to 17%, correct would be rounded 16%. Figure 1 shows 15 administrations of infliximab but only 10 infusions, since infliximab can only be given i.v., this is not comprehensible.

In summary, all calculations should be checked and corrected if necessary, as I can only check the obviously verifiable calculations.

The linguistic quality was not improved in a way that would allow barrier-free reading. A native-language revision could increase the informative value.

Author Response

Dear Reviewer 2,

Thank you for your comments and extensive review. We hope we succeded in adjusting the manuscript to your satisfaction.

  • Some inaccuracies regarding diagnoses and classifications could be reduced, but the relevant limitations remain. unfortunately, the linguistic quality was not improved in a way that would allow barrier-free reading. A native-language revision could increase the informative value.

We revised and re-worded the entire manuscript according to the best of our abilities, in consultation with our expert in English. We hope in this form it allows barrier free reading.

  • I cannot understand some of the changes: On page 3, line 147, the percentage of women was shortened from 69.85 to 67%, although with 146 women and 209 patients the percentage remained the same, correct would be 70%. The percentage of TNF prescriptions (page 4 line 169) was increased from 61.47 to 62%, rounded 61% would be correct. In line 172, the proportion of IL17 prescriptions was increased from 16.47 to 17%, correct would be rounded 16%. Figure 1 shows 15 administrations of infliximab but only 10 infusions, since infliximab can only be given i.v., this is not comprehensible.

In summary, all calculations should be checked and corrected if necessary, as I can only check the obviously verifiable calculations.

We apologise, all typos and miscalculations were corrected. We sincerely apologise for the confusion, as we had some technical issues with our database some numbers were pooled incorrectly. We took the time and checked and corrected everything manually. As you can see, there are no significant changes apart from intravenous use – the problem was precisely in infliximab. As infliximab was manufactured for subcutanous use in COVID-19 our system recorded the use of infliximab as partly subcutanous, but, as it was evident from medical documentation and the inventory, no infliximab was administered subcutaneously as it was unavailable and not part of the inventory at the time. To be certain, we manually checked all the remaining data and were assured of their accuracy.

Reviewer 3 Report

unfortunately, I still do not find a connection between changes in the use of a biological drug and the pandemic period, apart from problems with the availability of tocilizumab. The authors still did not refer to the specific reasons for the treatment change: intolerance, lack/loss of effectiveness, side effects. The presented study refers to the sales trends of various drugs over the last three years, related to, for example, the availability of new therapeutic options, but not to the pandemic period. The title of the article should reflect this.

Author Response

Response to the Reviewer 3

unfortunately, I still do not find a connection between changes in the use of a biological drug and the pandemic period, apart from problems with the availability of tocilizumab. The authors still did not refer to the specific reasons for the treatment change: intolerance, lack/loss of effectiveness, side effects. The presented study refers to the sales trends of various drugs over the last three years, related to, for example, the availability of new therapeutic options, but not to the pandemic period. The title of the article should reflect this.

Dear Reviewer 3,

Thank you for taking the time to evaluate the revised manuscript. As we are unable to provide sufficient clinical data we updated the title and the aim of our study, we hope to your satisfaction. They now state:

Response to the Reviewer 3

unfortunately, I still do not find a connection between changes in the use of a biological drug and the pandemic period, apart from problems with the availability of tocilizumab. The authors still did not refer to the specific reasons for the treatment change: intolerance, lack/loss of effectiveness, side effects. The presented study refers to the sales trends of various drugs over the last three years, related to, for example, the availability of new therapeutic options, but not to the pandemic period. The title of the article should reflect this.

Dear Reviewer 3,

Thank you for taking the time to evaluate the revised manuscript. As we are unable to provide sufficient clinical data we updated the title and the aim of our study, we hope to your satisfaction. They now state:

„Prescription trends of biologic DMARDs in treating rheumatologic diseases – changes of medication availability in COVID-19“

„Therefore, the aim of this study was to determine how the COVID -19 pandemic has af-fected the prescription of biologic DMARDs for rheumatologic diseases at the University Hospital Split, to identify the market trends of specific drugs, and to discuss how this might affect the quality of care.“

Reviewer 4 Report

The authors partially risponded to requests. Unfortunately there is a need for further major corrections:

 1)      The clinical data are needed to are needed to motivate the change. In case of lack of this data the authors should change the objectives of the study and better explain the description of the study (for example, description of the commercial trend of the drug) and correct the title of the study

2)      All calculations should be revised

Author Response

Response to the Reviewer 4

Dear Reviewer 4,

Thank you for taking the time to review our revised manuscript. We hope that in our

The authors partially risponded to requests. Unfortunately there is a need for further major corrections:

  • The clinical data are needed to are needed to motivate the change. In case of lack of this data the authors should change the objectives of the study and better explain the description of the study (for example, description of the commercial trend of the drug) and correct the title of the study

As we are unable to provide sufficient clinical data we updated the title and the aim of our study, we hope to your satisfaction. It now states:

„Prescription trends of biologic DMARDs in treating rheumatologic diseases – changes of medication availability in COVID-19“

„By studying these changes we can analyze how has potentially medication shortage, lack of face-to-face visits, physician availability, and number of referred patients affected the care and treatment of these patients. Therefore, the aim of this study was to determine how the COVID-19 pandemic has affected the prescription of biologic DMARDs for rheumato-logic diseases at the University Hospital Split, to identify the market trends of specific drugs, and to discuss how this might affect the quality of care.“

2)      All calculations should be revised

We apologise, all typos and miscalculations were corrected. We sincerely apologise for the confusion, as we had some technical issues with our database some numbers were pooled incorrectly. We took the time and checked and corrected everything manually. As you can see, there are no significant changes apart from intravenous use – the problem was precisely in infliximab. As infliximab was manufactured for subcutanous use in COVID-19 our system recorded the use of infliximab as partly subcutanous, but, as it was evident from medical documentation and the inventory, no infliximab was administered subcutaneously as it was unavailable and not part of the inventory at the time. To be certain, we manually checked all the remaining data and were assured of their accuracy.
